# Development and pilot of a multicriteria decision analysis (MCDA) tool for health services administrators

Robin Blythe,[1] Shamesh Naidoo,[2] Cameron Abbott,[2] Geoffrey Bryant,[2] Amanda Dines,[2] Nicholas Graves[1]

¹Australian Centre for Health Services Innovation, Institute of Health and Biomedical Innovation, School of Public Health and Social Work, Queensland University of Technology, Brisbane, Queensland, Australia
²Administration, Royal Brisbane and Women's Hospital, Herston, Queensland, Australia

**Correspondence to**
Robin Blythe;
robin.blythe@qut.edu.au

## ABSTRACT

**Introduction** Health administration is complex and serves many masters. Value, quality, infrastructure and reimbursement are just a sample of the competing interests influencing executive decision-making. This creates a need for decision processes that are rational and holistic.

**Methods** We created a multicriteria decision analysis tool to evaluate six fields of healthcare provision: return on investment, capacity, outcomes, safety, training and risk. The tool was designed for prospective use, at the beginning of each funding round for competing projects. Administrators were asked to rank their criteria in order of preference. Each field was assigned a representative weight determined from the rankings. Project data were then entered into the tool for each of the six fields. The score for each field was scaled as a proportion of the highest scoring project, then weighted by preference. We then plotted findings on a cost-effectiveness plane. The project was piloted and developed over successive uses by the hospital's executive board.

**Results** Twelve projects competing for funding at the Royal Brisbane and Women's Hospital were scored by the tool. It created a priority ranking for each initiative based on the weights assigned to each field by the executive board. Projects were plotted on a cost-effectiveness plane with score as the x-axis and cost of implementation as the y-axis. Projects to the bottom right were considered dominant over projects above and to the left, indicating that they provided greater benefit at a lower cost. Projects below the x-axis were cost-saving and recommended provided they did not harm patients. All remaining projects above the x-axis were then recommended in order of lowest to highest cost-per-point scored.

**Conclusion** This tool provides a transparent, objective method of decision analysis using accessible software. It would serve health services delivery organisations that seek to achieve value in healthcare.

## INTRODUCTION
### Background

Health services have a complex decision-making environment and there are many opportunities for innovation.[1] As identifying high value in healthcare becomes increasingly important, administrators must choose between the many ideas presented to them.[2–4]

### Strengths and limitations of this study

► This paper provides a concise and accessible method of multicriteria decision analysis (MCDA) for health services research.
► The tool described in this paper has been extensively field tested to fit with organisational goals of a large teaching hospital.
► There are many ways of conducting MCDA, with many potential criteria and weighting methods. The criteria decided on were bespoke for the associated hospital and may not confer full external validity to other organisations.
► Some fields, such as outcomes, depend on a literature review. There exists the possibility of bias in populating the tool, and we recommend an impartial party to administer the scoring.

Different initiatives such as new diagnostic hardware and service redesign compete for limited funds, but objectively choosing between initiatives is challenging.[5] The preferences of multiple stakeholders need to be accounted for, including payers, clinicians and patients, who all influence healthcare provision.[1]

While payers desire value, other stakeholders have heterogeneous preferences. Physicians have the best interests of the patient in mind, but can cause conflict with hospital cost containment measures when preference items such as prosthetics vary substantially in price.[6] Patient preferences may often directly oppose both cost-containment and physician preferences in an effort to obtain what patients perceive as optimal care. 'Doctor shopping' is one such example where patients will often circumvent value-based care by seeing multiple providers until they get the treatment they want.[7] However, not all patient and physician preferences impact negatively on value-based care.

## Rationale

Innovations to health services might provide benefits in many ways, yet these may not be comparable across projects. For example, a project to reduce medication mismanagement is not directly comparable to investing in an imaging device, yet these initiatives may find themselves in direct competition for funding. Considering every decision as an economic analysis is one method of evaluating projects using the same criteria, but this approach has been considered contentious and simplistic.[8–10] Multiple perspectives often need to be represented and this creates a need for an objective multicriteria decision analysis (MCDA) framework.

MCDA is a method of evaluating the performance and relative importance of different adoption decisions in a holistic manner. MCDA tool has two aspects: quantifying performance and quantifying the weight of each performance category on the overall decision.[11 12] This enables direct comparison across competing projects with multiple objectives.[13] MCDA can provide decision-makers with visual representations of their preferences and how to prioritise high value care.[12 13]

The aim for this paper is to describe a simple and transparent MCDA framework for competing projects. It was developed in partnership with the leadership of a 900 bed public teaching hospital in Brisbane, Australia. The hospital employs more than 6000 staff, admits over 100 000 patients each year and is heavily involved in research. The hospital's executive board used the decision tool over multiple iterations to improve its applicability and relevance. The value of the tool is that it is simple to use, employs a transparent methodology and provides visual summary of the outcomes to aid comparison and decision-making. We also provide some examples and outputs that show what the tool can do and how it can be used in the health system decision-making process.

## Methods

Values for six parameters are required for input into the MCDA tool, which uses Microsoft Excel 2016. The six criteria evolved from a set of priorities determined by the Royal Brisbane Women's Hospital (RBWH), the Brisbane Metro North Hospital and Health Service and recommendations from the Agency for Healthcare Research and Quality.[14]

### Patient and public involvement

No patients were recruited for this paper.

### Six parameters used for scoring projects

The net cost of implementing the programme is measured against the total benefits of its implementation. This represents the accounting cost to the provider of running the programme, rather than the opportunity cost of one programme over another.

### Cash return on investment

Cash return on cash investment (ROI) is expressed as a ratio of dollars returned for every dollar invested. We

**Table 1** Calculating return on investment (ROI)

| ROI equation | $\left(\dfrac{Profit}{Cost}\right) = ROI$ |
|---|---|
| Example | $\left(\dfrac{\$1,500,000}{\$5,000,000}\right) = 0.3$ |

used a linear transformation to keep scores between 0 and 2. The highest ROI is used as the denominator and given a score of 2, while each subsequent ROI is scored as a proportion of the maximum and multiplied by 2. Calculating ROI is shown in table 1. This field calculates returns in terms of cash only. Any financial benefits gained by improvements to capacity or safety, such as bed days avoided or lower medical costs from falls reductions, must not be included.

For example, if a project's 5-year profit was $1 500 000 and cost $5 000 000, its ROI would be ($1 500 000/$5 000 000)=0.3. If the highest ROI was 4.0, this would receive a score of 2×(0.3/4.0)=0.15.

### Capacity changes

The second parameter is whether the innovation releases capacity. These costs are prepaid or fixed in that the funding has already been allocated to keeping beds open regardless of occupancy, and increasing bed availability will not free up any cash.[15] We developed the 'c-score' to measure change to capacity in terms of the length of stay (LOS) of the relevant clinical unit. The c-score is outlined in table 2. This formula adds the total change in throughput (change in LOS×number of patients) to the weighted impact of the added capacity (bed days added/LOS). This allows flexibility in how capacity is measured to include both changes to LOS and the impact of available beds in different wards. The weighted bed days component reflects the relative impact of freeing 24 hours of Emergency Department (ED) time compared with 24 hours of a standard ward. This field also uses linear transformation as per the ROI score.

For example, a project reduces LOS by 3 hours (0.125 days), affecting 40 patients per week on average. Average LOS is 5 hours, but no beds have been added, meaning the total c-score is (0.125*40)=5. This is the maximum c-score of any project and receives a score of 2× (5/5)=2.

### Patient benefits

Scores to signal improvements to quality of life and patient satisfaction vary from 0 to 2. A score of 2 is assigned for quality of life when the literature shows,

**Table 2** Calculating the c-score

| C-score equation | $\left(\Delta\ LOS\ in\ days \times patients\ per\ week\right) + \dfrac{Added\ bed\ days}{LOS\ of\ unit\ in\ days}$ |
|---|---|
| Example | $\left(0.125\ days \times 40\right) + \dfrac{0}{5\ days} = 5$ |

LOS, length of stay.

through meta-analysis or multisite Randomised Control Trial (RCT), improvements in health utility or health-related quality of life. A score of 1 would be given to projects with no available or reputable evidence in the literature, and a score of 0 for projects which have clear evidence of harm or ineffectiveness. Patient satisfaction must follow the same criteria. Access, defined as a reduction in waiting or travel time and patient costs, is a binary 0 or 1 for 'no improvement' or 'improvement', respectively, as measured in meta-analysis or primary data collection tools such as the Research and Development Corporation (RAND) Patient Satisfaction Questionnaire (PSQ)-18.

Due to the potential for bias, it is important that objective parties conduct a brief transparent and reproducible literature review to provide sound judgement. Advocates may overlook negative findings and focus on positive ones. As with the ROI and capacity fields, any purported improvements to quality of life must not arise from improvements in patient safety or reductions in length of stay, to avoid multicollinearity. An initiative with an evidence basis for all three parameters would receive a maximum score of 5, then scaled by the linear transformation method used for ROI and capacity. The score breakdowns are shown in table 3.

For example, a programme to install laminar airflow devices in all operating theatres displayed evidence in the literature of no improvement and, in some cases, even harm, scoring a 0 on quality of life. Patients were unaware of the development and did not see reduced wait times, scoring a 0 on both satisfaction and access. The initiative scored 0. The project with the highest outcomes scored 4, so the laminar airflow project was given a score of 2× (0/4)=0.

## Patient safety

Patient safety indicators for Australian hospitals arise from the Australian Commission on Safety and Quality in Health Care which monitors 16 hospital-acquired conditions.[16] The number of indicators addressed is summed and linearly transformed as per the above criteria. As with outcomes, there must be a clear body of evidence, either on balance or through meta-analysis, showing an improvement in the reported safety outcomes. For example, a project that incorporated daily, assisted walks with long stay patients addressed three conditions, including

pressure ulcers, falls and venous thromboembolisms. The project addressing the most conditions scored a 6, giving the assisted walks project a 2× (3/6)=1.

## Staff training and research

Staff training and research is required for many clinical personnel and represents an important part of professional development. It can take the form of sanctioned continuing professional development hours, scientific publication or simply improved skill sets and job satisfaction. Table 4 outlines the breakdown of training and research outcomes. These scoring outcomes must be determined by an objective party. As the field is scored from 0 to 2, no linear transformation is necessary.

For example, a project resulted in an academic publication qualifying it as research, but provided no additional training, scoring 1/2.

## Organisational risk

Organisational risk weighs the impact of potential risks against the probability of their occurrence. Examples of possible risks include delayed implementation or a negative news article around the project. Classifying impact and probability depends on the organisation's priorities and characteristics. For example, an organisation may deem probabilities below 10% as low and above 60% as high, and any project requiring debt financing as high impact. The intersection of risk impact and probability creates a matrix with a scoring system from 0 to 2. Ideal projects would have both a low impact and low probability of occurrence, scoring a 2. As with training and research, no scaling factor is required. Figure 1 displays the risk classification matrix.

For example, a project was estimated to have a 1 in 3 chance of stalling when its main sponsor went on long

| Table 4 Scoring for training and research | |
|---|---|
| **Training/Research provided** | **Score** |
| No training or research provided through project | 0 |
| Provides training or research | 1 |
| Provides training and research | 2 |

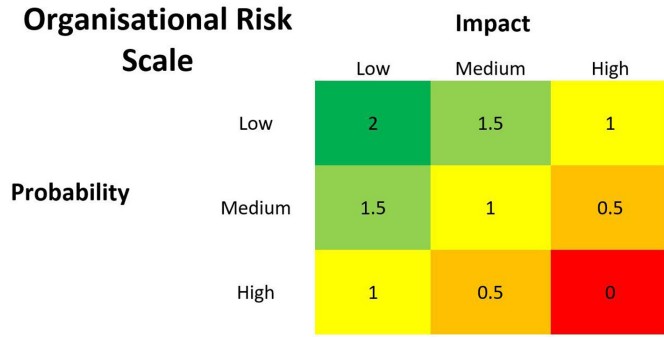

**Figure 1** Organisational risk scale for impact and probability assessments.

| Table 3 Scoring patient-related outcome measures | | | | |
|---|---|---|---|---|
| **Component** | **Score=0** | **Score=1** | **Score=2** | **Range** |
| Quality of life | Evidence of harm/no improvement | Evidence base uncertain | Evidence of improvement | 0–2 |
| Patient satisfaction | Evidence of harm/no improvement | Evidence base uncertain | Evidence of improvement | 0–2 |
| Access to care | Does not improve access | Improves access | | 0–1 |
| Total | | | | 0–5 |

service leave, classified by the board as a medium probability, low impact risk and scoring 1.5/2.

## Weighting and discounting for outcomes in future time periods

Discounting is a standard procedure in financial valuation to account for time preference.[17] We used the widely accepted 3% discount rate for ROI. Weighting is a more complicated system of establishing the priorities of the hospital decision-makers.[11 17] Each value is multiplied by a weight and summed for the total project score. It is important that weights are agreed collaboratively ahead of time to account for heterogeneous preferences and to recognise the needs of different groups.[18]

We created weights by asking executives to rank fields by priority, in which two priorities could be of equal rank. Table 5 below shows a ranking system in which decision-makers prioritised ROI and outcomes, followed by safety, capacity, then training and risk as joint bottom. Ranks were summed, with the total then divided by each ranking for a score.

Maintaining consistent weights across a single decision window is crucial for objective measurement. Administrators and executives should decide on priorities and only change them once funding has been allocated and a new decision time period has begun. If weights are determined during or after projects are measured by the decision framework, it will introduce bias towards initiatives offering more points in the selected criteria. Rankings should be achieved through a discussion among key decision-makers and left unchanged for the remainder of the funding window.

## RESULTS

Using the methods detailed above, the tool analysed two projects competing for funding at the RBWH: early patient intervention centre (EPICentre, an accelerated triage unit, and the Elective Surgery (ES) Pod for low-risk surgical recovery.

**Table 5** Rankings determined by executives prior to analysis

| Field | Rank | Score=Total/ Rank | Weight=Score/ Total |
|---|---|---|---|
| ROI | 1 | 15 | 0.3 |
| Capacity | 3 | 5 | 0.1 |
| Outcomes | 1 | 15 | 0.3 |
| Safety | 2 | 7.5 | 0.15 |
| Training | 4 | 3.75 | 0.075 |
| Risk | 4 | 3.75 | 0.075 |
| Total | 15 | 50 | 1 |

ROI, return on invesment.

### Return on investment

EPICentre costs a flat $1.1 million each year, with no cash returned and an ROI of 0. ES Pod costs $3.75 m over 5 years, and generates $4.91 m in that time. ES Pod's ROI after 5 years is $1.15 m divided by its cost of $3.75 m, or 0.31. Scaled as per the methods to the highest scoring project with an ROI of 1.21, ES Pod scored 2 (0.31/1.21), or 0.51 out of 2.

### Capacity

EPICentre decreases ward LOS by 5 hours by reducing wait times for specialist consults. The project affects 39.3 patients per week, but adds no beds. Its c-score is (39.3×0.22+0)=8.6. Scaled to the highest scoring project of 35.7, it scored 2 (8.6/35.7), or 0.48 out of 2. ES Pod does not decrease LOS, but allows 3.4 extra patients to be admitted to the intensive care unit (ICU) per week, where the average LOS is 3.3 days. The c-score for ES Pod is (0+3.4/3.3)=1.03, scaled to 2× (1.03/35.7) or 0.06 out of 2.

### Outcomes

There is no evidence of EPICentre improving patient quality of life outcomes directly, but a pilot survey showed increases in patient satisfaction and throughput, improving access. These outcomes summed to 4, leading all initiatives for a score of 2 out of 2. ES Pod shows no available evidence for changes to quality of life and satisfaction, but improves access through lower wait times. Outcomes summed to 3, receiving a score of 2× (3/4), or 1.5 out of 2.

### Safety

EPICentre was shown in a pilot programme to reduce falls, Hospital Acquired Infections (HAIs), unplanned ICU visits and venous thromboembolisms. The sum of four indicators was 1 short of the highest score of 5, calculated as 2× (4/5) or 1.6 out of 2. ES Pod was shown to be non-inferior to an ICU setting, but offered no additional safety benefits and received a score of 0/2.

### Training and research

EPICentre and ES Pod were two of the highest scoring projects, with both offering Continuing Professional Development (CPD) hours and research manuscripts, receiving scores of 2/2.

### Risk

Hospital executive board members determined that both EPICentre and ES Pod had medium probability risks (between 10% and 60%) of some financial risk and negative publicity, respectively. Both were allotted a risk score of 1.5/2.

### Weighting

Each outcome measure was weighted according to preset preference allocations as shown in table 5. The weighted scores of all competing projects can be seen in table 6, ranked by cost-per-point gained.

**Table 6** Total cost and impact scores on an individual project basis

| Project | 5year ROI | Capacity | Outcomes | Safety | Training | Risk | Net Cost | Score | Cost per point |
|---------|-----------|----------|----------|--------|----------|------|----------|-------|----------------|
| CF2 | 0.52 | 0.00 | 0.30 | 0.00 | 0.08 | 0.15 | -$7 190 877 | 1.05 | -$6 858 599 |
| APHS | 0.03 | 0.00 | 0.30 | 0.00 | 0.00 | 0.15 | -$982 496 | 0.48 | -$2 051 006 |
| ES Pod | 0.15 | 0.01 | 0.45 | 0.00 | 0.15 | 0.11 | -$1 155 429 | 0.87 | -$1 324 633 |
| FIM | 0.60 | 0.00 | 0.30 | 0.00 | 0.08 | 0.15 | -$1 290 202 | 1.13 | -$1 146 846 |
| CN/CF | 0.00 | 0.00 | 0.30 | 0.00 | 0.15 | 0.15 | $0 | 0.60 | $0 |
| OPAT | 0.00 | 0.00 | 0.45 | 0.00 | 0.08 | 0.15 | $501 616 | 0.68 | $742 494 |
| SW | 0.00 | 0.19 | 0.30 | 0.24 | 0.00 | 0.15 | $1 103 631 | 0.88 | $1 247 107 |
| TMT | 0.00 | 0.20 | 0.45 | 0.12 | 0.08 | 0.15 | $1 375 846 | 1.00 | $1 382 759 |
| VASE | 0.00 | 0.00 | 0.30 | 0.06 | 0.08 | 0.04 | $1 815 965 | 0.47 | $3 843 312 |
| Eat Walk Engage | 0.00 | 0.00 | 0.60 | 0.30 | 0.08 | 0.15 | $5 055 734 | 1.13 | $4 493 986 |
| EPICentre | 0.00 | 0.05 | 0.60 | 0.24 | 0.15 | 0.11 | $5 283 150 | 1.15 | $4 590 555 |
| CEP-CARU | 0.00 | 0.00 | 0.45 | 0.00 | 0.15 | 0.00 | $10 476 370 | 0.60 | $17 460 617 |

APHS, insourcing chemotherapy supply; CEP-CARU, epilepsy surgical centre; CF2, chemotherapy funding change; CN/CF, clinical nurse/clinical facilitator; EPICentre, early patient intervention centre; ES Pod, elective surgery pod; FIM, functional independence measurement; OPAT, outpatient antibiotic therapy; SW, social worker support; TMT, tracheotomy management team; VASE, vascular access education.

A rational approach is to maximise the number of points obtained for a given budget by accepting all cost-saving and no cost projects (Chemotherapy funding change (CF2), insourcing chemotherapy supply (APHS), ES Pod, functional independence measurement (FIM), clinical nurse/clinical facilitator (CN/CF)), then selecting projects with the cheapest cost-per-point until the budget is exhausted. Figure 2 displays projects with a positive cost-per-point where the x-axis is points gained and the y-axis is cost.

Dominated projects are those such as vascular access education (VASE) and epilepsy surgical centre

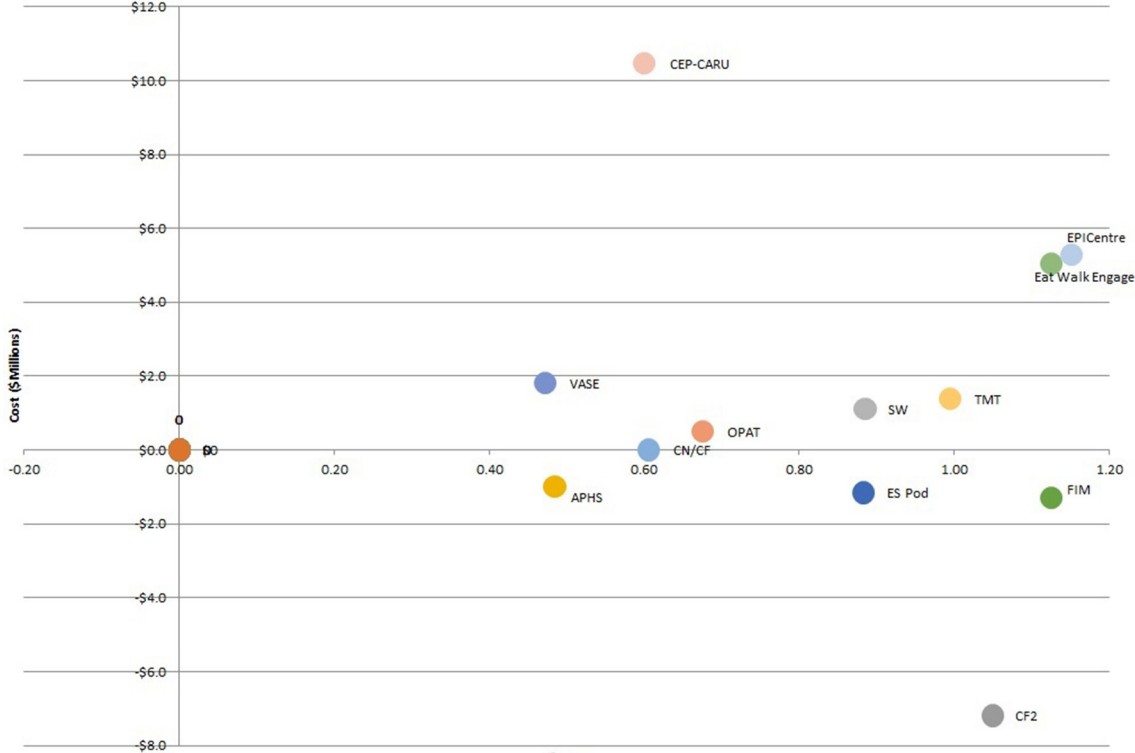

**Figure 2** Plotting projects on a plane after cost-saving projects are accepted. APHS, insourcing chemotherapy supply; CEP-CARU, epilepsy surgical centre; CF2, chemotherapy funding change; CN/CF, clinical nurse/clinical facilitator; EPICentre, early patient intervention centre; ES Pod, elective surgery pod; FIM, functional independence measurement; OPAT, outpatient antibiotic therapy; SW, social worker support; TMT, tracheotomy management team; VASE, vascular access education.

(CEP-CARU) which are both less effective and more expensive than an alternative, and should not be pursued before the dominant strategies. Projects on the cost-effectiveness frontier, such as outpatient antbiotic therapy (OPAT), social worker support (SW), and tracheotomy management team (TMT) should be pursued first, in that order. This choice provides a good example of cost-effectiveness decision-making as there is a clear trade-off between points and costs as projects move up and right. A rational decision-maker might have a budget of $10m for implementation and could use the decision tool to choose projects with the best cost-per-point starting with OPAT until the $10m was depleted.

## DISCUSSION

A hospital executive committee could use this tool to sanction purchases for each funding window. A health system funding an innovation grant may use it to judge the relative merits of different applications. In any case, it should complement a broader implementation decision, using other data such as organisational readiness for change.

While the MCDA tool was built for prospective projects, it is also capable of retrospective decision-making. Data entered will have the advantage of being based on observations rather than conjecture. The reason for retrospectively evaluating projects can be to recognise successful new models of care, but it can also be to disinvest. This is a highly contentious issue in health service delivery, but as cost pressures increase, health systems may find funding for new projects only by phasing out older, ineffective ones.[2] There are commonly costly and ineffective policies that could be phased out, but may require explicit demonstration relative to other projects.[19] This tool can provide the rationale for disinvestment in projects occupying the top left quadrant, allowing funds to be better spent in the high-value sections. It can also help determine where to invest or disinvest in situations where financial constraints force an organisation to reduce benefits in order to cut costs.

It is important to avoid multicollinearity across fields. An important distinction between capacity and ROI is that the latter is a cash-only metric. An initiative that frees up cash directly by generating additional revenues or cost savings is not the same as one that frees up a similar amount of beds. Capacity informs key benchmarks, such as the National Emergency Access Target known in the UK as the '4-Hour-Rule,' and both loses nuance and requires tacit valuations when reported as a financial figure.[20] Similarly, an initiative reducing falls should be scored by patient safety, rather than by the outcome improvement in health utility or capacity gains from reduced LOS that might accompany a reduction in falls.

## Implementation at the RBWH

The RBWH established a team in 2016 to retrospectively and prospectively evaluate service improvements at various stages of implementation. Needing a transparent and simple method of evaluation, the RBWH initially adopted an evaluation framework already in use at the regional level. This method resulted in large reports that consumed significant amounts of staff time in reporting, requiring significant drafting and editing to convey findings.

The methods used in this paper offered the RBWH a more targeted and concise approach to evaluation. While some written reporting is still necessary to summarise the implementation history and effectiveness of each initiative, adoption of this interface allows the RBWH to perform retrospective and prospective analyses in a shorter timeframe with more concise findings. This facilitates faster funding allocation decisions across competing programmes.

The RBWH has begun using the multicriteria decision-making tool to inform future resource allocation. The use of this decision-making tool gives decision-makers and funding applicants a level of confidence that allocations have been made in a systematic and deliberate manner. This predefined, objective set of criteria allows fair discrimination across competing projects.

### Limitations

There are some drawbacks to this approach over non-MCDA methods. A notable downside is that political influences, such as publicly reported deadlines and government pressure, are not quantifiable within the tool. These factors tend to transcend typical decision metrics and distort objective, scientifically supported choices.[21] Political pressures may also be reflected in the weighting of different criteria. Occasionally there are situations where a project must be completed for political reasons rather than for the benefit granted to the health system. In cases like this, the tool does not provide a justification for the decision to invest, but instead quantifies impacts of the project and how they compare to other interventions. By plotting politically necessitated projects on the graph in figure 2, it is also possible to show project outcomes to the political bodies demanding certain initiatives, and show why they may or may not be an effective option.

This tool is intended to be a guide, not a mandate, and will not solve healthcare problems overnight. It is also only as good as the data that go into it; gaming through selective interpretation of findings could misrepresent the nature of an intervention, twisting the tool by adding a veneer of credibility to subjective opinions. Similarly, avoiding double-counting is explicitly addressed in the methods, but a potential pitfall that must be avoided by the analyst. MCDA is just a component of an executive decision-making process, to be included in a holistic review of options. Ideally, it would be conducted at the beginning of the funding window as a prospective analysis of the expected benefits of all initiatives competing for funding. This is applicable to hospitals, health systems and hospital-based public policy, as MCDA can be valuable in all of these sectors.[5]

While many projects are cost-saving, the time horizon for these savings may accrue after several funding windows or years. This may mean that a project will not break even for 4 years, before a large windfall in the fifth. This nuance can be captured through a time factor in the ROI scoring system, where the interests of decision-makers can help choose the year to apply. Similarly, cost-saving projects also may require a significant implementation burden, which can often go beyond the system's administrative capacity. If a ward or unit only has the staff time to implement four projects of similar scope per year, then funding is not as large a constraint as time and capacity. This highlights the use of the decision tool as part of a larger toolkit. Some decisions are beyond purely objective or quantifiable criteria, and require holistic analysis with input from all parties.

Weighting is determined in isolation from project performance. This removes the impact of a change's magnitude on the weighting factor, such as might be seen in swing weighting. Instead, weights and performance are synthesised after they have been individually considered. This has some downsides. For example, a major improvement in health outcomes with a slight amount of risk may be scored above a slight improvement in health outcomes with no risk. We have attempted to address this in part through the relative nature of the linear transformation scale. When this does not apply, however, we ultimately must stress that this tool cannot be a full cost-effectiveness analysis on outcomes. As above, it must be considered as a component of an overall decision process, in which cost-effectiveness and equity should play a large part.

Finally, due to the subjective nature of weighting and some of the scoring metrics, we had to sacrifice some scientific rigour for ease of use and transparency. Ideally, each field would have had a methodologically robust scoring system in which each outcome could be objectively defined and multicollinearity avoided through explicit value definitions. Due to the prospective design, short timeframes and need for transparent and easily understood scoring process, true precision was difficult to achieve. Under the circumstances, we believe this tool strikes an acceptable balance between rigour and convenience.

## CONCLUSIONS

This MCDA tool is flexible with weighting, allowing different outcomes to be prioritised based on heterogeneous preferences and political pressures. It provides a way of comparing otherwise incomparable outcomes. It is user-friendly and requires no additional technology in an office setting, giving it broad applicability. It is transparent, where the benefits and costs of different projects have a clear accounting method that can be challenged through review and debate.

This has several implications for hospitals and health systems. By selecting projects using MCDA, funding can be allocated in the most efficient manner. Projects with

negative outcomes can be identified before they can negatively affect the health system. The tool can also analyse politically motivated projects and provide a basis of comparison that can explicitly address subjective preferences. By comparing each initiative on equal footing, there is a lower chance for bias to affect systems-level policy and a higher chance for projects with genuine benefit to be funded and implemented.

**Contributors** RB: Conceptualisation, analysis, investigation, methodology, visualisation, writing, review. SN: Conceptualisation, analysis, investigation, methodology, visualisation, writing, review. CA: Analysis, investigation, methodology, review. GB: Conceptualisation, methodology, visualisation. AD: Conceptualisation, writing, review. NG: Conceptualisation, analysis, investigation, methodology, writing, review.

**Funding** The authors have not declared a specific grant for this research from any funding agency in the public, commercial or not-for-profit sectors.

**Competing interests** None declared.

**Patient consent for publication** Not required.

**Provenance and peer review** Not commissioned; externally peer reviewed.

**Data sharing statement** Data on the specific projects evaluated in the tool have been obtained from the RBWH hospital in Brisbane, Australia. If desired, please contact the RBWH for further information.

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
