## [Reviewer comments · BMJ Open]

This paper was submitted to a another journal from BMJ but declined for publication following peer review. The authors addressed the reviewers' comments and submitted the revised paper to BMJ Open. The paper was subsequently accepted for publication at BMJ Open.

(This paper received three reviews from its previous journal but only two reviewers agreed to published their review.)

ARTICLE DETAILS

TITLE (PROVISIONAL)	Development and pilot of a multi-criteria decision analysis (MCDA) tool for health services administrators
AUTHORS	Blythe, Robin; Naidoo, Shamesh; Abbott, Cameron; Bryant, Geoffrey; Dines, Amanda; Graves, Nicholas

VERSION 1 – REVIEW

REVIEWER	Jesper Bo Nielsen Department of Public Health, University of Southern Denmark, Denmark
REVIEW RETURNED	07-Sep-2018

GENERAL COMMENTS	The transparency offered by a MCDA approach seems very relevant also for administrative decisions and prioritizations. Thus, the topic of this manuscript is clearly interesting and relevant. However, the underlying premise and principle for MCDA is that ratings of the chosen criteria are totally separated from the weightings of the criteria. Major issue: The authors have made the fundamental error of giving three of their criteria max values of 5 and the other three a max value of 2. By doing this, the authors have essentially put in their personal weightings on those three criteria. The reason is that the basic algorithm used for calculating the scores are additive, which means that those criteria with a max value of 5 will for mathematical reasons be weighed 2½ times higher than the other. In this way, the authors are basically corrupting the subsequent weighing process, and even in a way that is not transparent to the user. In the section on limitations, the authors address this issue by stating that they find the different max values OK because they find pre-weighing OK. This reviewer does not agree. Any weighing should be done transparently during the weighing process, where the users have all possibilities to weigh some criteria above others. The idea of using the MCDA in choosing between new initiatives is relevant, and the authors should be encouraged to change the rating scales so that they are identical across criteria (they could essentially just divide all ratings from criteria one, two, and three by 2½ before entering them into the score calculation). But as they are now, they cannot be used. Minor issues:
--

	In Table 1 as well as in the Result section under subheading ROI, there appears to be two miscalculations.
--	--

REVIEWER	Maureen Rutten-van Molken Erasmus University Rotterdam, the Netherlands
REVIEW RETURNED	19-Sep-2018

GENERAL COMMENTS	This paper presents a framework to perform a value-based MCDA of hospital project-investment decisions. It is applied to prioritize 12 hypothetical projects on the basis of 6 criteria. The performance of the projects on these 6 criteria is scored and subsequently weighted and the partial weighted value scores are summed to obtain the overall value score. The decision criteria are all relevant and the framework is analytically simple and transparent, which may benefit its use in practice. However this simplicity comes at the price of scientific rigor. A number of conditions that need to be met to obtain valid results are violated. I recommend the authors to pay much more attention to this in the discussion, under 'limitations'. According to multi-attribute value theory, the 6 decision criteria need to be preferentially independent, i.e. the weight of one criterion should be independent of the performance on other criteria. That is clearly not the case because for example the weight assigned to patient benefits (which includes patient satisfaction) is likely to depend on the performance on safety, and the weight assigned to ROI is likely to depend on the reduction in LOS. Some of the scales on which the criteria are measured are rather arbitrarily defined. For example, how is 'clear evidence' of improvement in patient benefits from a 'good' article defined? Did you use any of the existing grading systems to score the level of evidence? Another example: measuring 1-2 indicators of patient safety gets a performance score of 1 whereas measuring 3+ indicators gets a performance score of 2. However, if a single criterion that is measured is of much more relevance than 3 minor criteria, why should the project that addressed 1 criterion be performing worse? Some scales are a composites of very different subcriteria (e.g. patient benefits includes QoL and access to care, both of which might be weighted very differently). It is generally acknowledged that criteria which are defined in terms of 'change' are hard to interpret if the starting value is not giving. That is why many MCDAs measure the performance in absolute scores. The same improvement in QoL at the lower end of a scale may be much more valuable than at the upper end of the scale. Several of the criteria are defined in terms of change (e.g. LOS, the 3 patient benefits, staff skills) without any indication of the starting level. Because the performance scores of the different criteria are measured on different natural scales, they have to be standardized into the same scale to make them comparable. The authors are using very arbitrary ways to (more or less) 'standardize' the first 3 criteria on a scale from 0-5 and the last 3 criteria on a scale from 0-2. For example what is the justification to multiply the ratio (profit/cost) with 10? According to MCDA theory one would either use 'relative standardization' of 'ranging
---

	standardization'. When using ranging standardization one can use either the local or global range. Why did the authors deviate from this common approach? How would the results be affected by a different method of standardization? The theoretically best-founded weights are those based on techniques that take account of the entire potential range of performance. Furthermore, weight-elicitation methods are preferred that force stakeholders to trade criteria off against one-another. Hence, the best methods to obtain relative weights are Swing Weighting and Discrete Choice Experiments. However, the authors assigned weights by dividing 100% over the 6 criteria. What was the justification for this method and how sensitive are the results to changes in these weights? I agree that the weights should not change within a certain investment time-window. There is a lot of debate about whether or not to include (net) costs in the set of decision criteria. Your approach in which you calculate the net costs per point increase in your new composite benefit score is theoretically preferred. It would be relevant to discuss that these costs do not represent opportunity costs, but costs from a provider (i.e. hospital) perspective. Furthermore, I would add a paragraph to the discussion in which you discuss the absence of a link between performance and weights. There is only one weight for each criteria, irrespective on where the change on the performance scale occurs. This has major limitations that are worth discussing. One of the limitations that you mention on page 2 is that the AHRQ were not suitable for hospital administrators. That does not seem to be a limitation of your own research and I don't understand how this conclusion would logically follow from your study, but that might be because I am not familiar with these criteria. Could you please clarify? I would recommend reading the papers of the ISPOR Task Force on MCDA by Marsh et al. plus a recent 'commentary' in Value in health 2018; 21: 394-397
--	--

REVIEWER	Jason C. Hsu National Cheng Kung University, Taiwan
REVIEW RETURNED	21-Sep-2018

GENERAL COMMENTS	 1. The main objective of the decision making of in this case (health service administration) should be addressed more specifically and clearly. 2. How were these criteria selected? Did the authors do the expert validity assessment for the present criteria? What is the relevance of the criteria to the main decision-making objective? Is there independence between the criteria and no collinearity? Are there any other criteria to reach the main objective which have not been included yet? 3. The weights of the criteria were set beforehand by decision bodies, which might be too subjective and not objective. 4. It is recommended that the author present the decision results in a "priority order" and explain how to apply the analysis results to subsequent actual decisions. 5. That would be better if the author can do the sensitivity analysis by the range of weights.
---

VERSION 1 – AUTHOR RESPONSE

Thank you to the reviewers for their thoughtful contributions.

In response to the reviews we have made the following changes where appropriate, listed below as bullet points.

Reviewer: 1

The transparency offered by a MCDA approach seems very relevant also for administrative decisions and prioritizations. Thus, the topic of this manuscript is clearly interesting and relevant.

However, the underlying premise and principle for MCDA is that ratings of the chosen criteria are totally separated from the weightings of the criteria.

Major issue:

The authors have made the fundamental error of giving three of their criteria max values of 5 and the other three a max value of 2. By doing this, the authors have essentially put in their personal weightings on those three criteria. The reason is that the basic algorithm used for calculating the scores are additive, which means that those criteria with a max value of 5 will for mathematical reasons be weighed 2½ times higher than the other.

In this way, the authors are basically corrupting the subsequent weighing process, and even in a way that is not transparent to the user.

- Inconsistent maximum values: In order to address this point, we scaled each factor out of two with a linear transformation. This transformation made each value a fraction of the maximum score received by any project. Each criteria is now out of two, satisfying this inconsistency issue. We believe this strikes a balance between convenience and rigour. This point is addressed in each paragraph of the methods, from pages 3-7.

In the section on limitations, the authors address this issue by stating that they find the different max values OK because they find pre-weighting OK. This reviewer does not agree. Any weighing should be done transparently during the weighing process, where the users have all possibilities to weigh some criteria above others.

- We agree with the reviewer's point, and this was the intention in the original manuscript. The decision body is encouraged to create weights based on priorities ahead of time, which are then static for the remainder of the process. This is specified in the methods on page 7, under 'Weighting and discounting for outcomes in future time periods'.

The idea of using the MCDA in choosing between new initiatives is relevant, and the authors should be encouraged to change the rating scales so that they are identical across criteria (they could essentially just divide all ratings from criteria one, two, and three by 2½ before entering them into the score calculation).

But as they are now, they cannot be used.

Minor issues:

In Table 1 as well as in the Result section under subheading ROI, there appears to be two miscalculations.

- As per the inconsistent values point above, we have taken the reviewer's advice on this matter. The typos in table 1, page 4 have been fixed as well.
- We thank the reviewer for their comments and agree with the changes suggested. These have been implemented in the paper.

Reviewer: 2

This paper presents a framework to perform a value-based MCDA of hospital project-investment decisions. It is applied to prioritize 12 hypothetical projects on the basis of 6 criteria. The performance of the projects on these 6 criteria is scored and subsequently weighted and the partial weighted value scores are summed to obtain the overall value score.

The decision criteria are all relevant and the framework is analytically simple and transparent, which may benefit its use in practice. However this simplicity comes at the price of scientific rigor. A number of conditions that need to be met to obtain valid results are violated. I recommend the authors to pay much more attention to this in the discussion, under 'limitations'.

- We have reinforced this point in the limitations section, page 10, 3rd paragraph.

According to multi-attribute value theory, the 6 decision criteria need to be preferentially independent, i.e. the weight of one criterion should be independent of the performance on other criteria. That is clearly not the case because for example the weight assigned to patient benefits (which includes patient satisfaction) is likely to depend on the performance on safety, and the weight assigned to ROI is likely to depend on the reduction in LOS.

- We find this a very valid critique. We have, as a result, specified on numerous occasions that only a single scoring item can be affected by a potential project outcome. We have attempted to reinforce this point, but ultimately needed to repeatedly stress this in the body text rather than change the decision metrics in the tool. This point is specified on page 4 under Cash ROI and Capacity changes, page 5 under Patient benefits, the first paragraph of page 8, and the final paragraph of Limitations on page 9.

Some of the scales on which the criteria are measured are rather arbitrarily defined. For example, how is 'clear evidence' of improvement in patient benefits from a 'good' article defined? Did you use any of the existing grading systems to score the level of evidence? Another example: measuring 1-2 indicators of patient safety gets a performance score of 1 whereas measuring 3+ indicators gets a performance score of 2. However, if a single criterion that is measured is of much more relevance than 3 minor criteria, why should the project that addressed 1 criterion be performing worse? Some scales are a composite of very different subcriteria (e.g. patient benefits includes QoL and access to care, both of which might be weighted very differently).

- The criteria for subjective measures, such as 'clear evidence' now explicitly requires proof-positive from a meta-analysis or multi-site RCT in the absence of scientific consensus. This is specified under patient benefits and patient safety, both on page 5.

It is generally acknowledged that criteria which are defined in terms of 'change' are hard to interpret if the starting value is not given. That is why many MCDAs measure the performance in absolute scores. The same improvement in QoL at the lower end of a scale may be much more valuable than at the upper end of the scale. Several of the criteria are defined in terms of change (e.g. LOS, the 3 patient benefits, staff skills) without any indication of the starting level.

- We have attempted to address this in two ways. First, the linear transformation scaling method mentioned above, detailed in pages 3-7. Second, while we acknowledge that starting value is a worthy consideration, we believed that a more utilitarian approach was necessary. Using the example of outcomes, two projects with a similar increase in health utility are typically considered equal, as this is the convention for current practice when making funding decisions around cost/QALY. Equity considerations revolving around the starting point of a population characteristic are typically political and such judgements tend to supercede objective decision frameworks. In terms of the relationship between weighting and performance, we hope that the scaling factor used in each criterion should address this issue more squarely.

Because the performance scores of the different criteria are measured on different natural scales, they have to be standardized into the same scale to make them comparable. The authors are using very arbitrary ways to (more or less) 'standardize' the first 3 criteria on a scale from 0-5 and the last 3 criteria on a scale from 0-2. For example what is the justification to multiply the ratio (profit/cost) with 10? According to MCDA theory one would either use 'relative standardization' or 'ranging standardization'. When using ranging standardization one can use either the local or global range. Why did the authors deviate from this common approach? How would the results be affected by a different method of standardization?

- We have attempted to address this in the 'inconsistent maximum values' point above, pages 3-7.

The theoretically best-founded weights are those based on techniques that take account of the entire potential range of performance. Furthermore, weight-elicitation methods are preferred that force stakeholders to trade criteria off against one-another. Hence, the best methods to obtain relative weights are Swing Weighting and Discrete Choice Experiments. However, the authors assigned weights by dividing 100% over the 6 criteria. What was the justification for this method and how sensitive are the results to changes in these weights? I agree that the weights should not change within a certain investment time-window.

- Weighting framework: We acknowledge that the weighting system from 0-100% for each criteria may be too subjective. We substituted a ranking system, or priority order, which seeks to address this algebraically, shown in table 5. Decision makers can now rate their priorities and the tool will return a value dependent on each rank as a function of the total ranking for each criterion. While less rigorous than a discrete choice experiment, it is more accessible. We did not conduct a sensitivity analysis, due to word count constraints, but believe that this system will better address unwanted variation from spontaneous changes in weights that may have arisen in the previous model. We have detailed these methods under 'Weighting and discounting for outcomes in future time periods' on pages 6 and 7.

There is a lot of debate about whether or not to include (net) costs in the set of decision criteria. Your approach in which you calculate the net costs per point increase in your new composite benefit score is theoretically preferred. It would be relevant to discuss that these costs do not represent opportunity costs, but costs from a provider (i.e. hospital) perspective.

- We have addressed this more succinctly in the methods section, page 4, in order to form a more compact definition of costs, especially relating to collinearity with capacity.

Furthermore, I would add a paragraph to the discussion in which you discuss the absence of a link between performance and weights. There is only one weight for each criteria, irrespective on where the change on the performance scale occurs. This has major limitations that are worth discussing.

- This has been added to the limitations section, page 9, paragraph 2.

One of the limitations that you mention on page 2 is that the AHRQ were not suitable for hospital administrators. That does not seem to be a limitation of your own research and I don't understand how this conclusion would logically follow from your study, but that might be because I am not familiar with these criteria. Could you please clarify?

- We initially formed the decision criteria upon discussion with executive board members of our hospital partner. We suggested the AHRQ criteria, which were interesting to the decision body, but ultimately deemed unsuitable to their internal processes without alterations. The final decision tool was inspired by the AHRQ but ultimately different. We have removed this segment from the paper to avoid confusion.

I would recommend reading the papers of the ISPOR Task Force on MCDA by Marsh et al. plus a recent 'commentary' in Value in health 2018; 21: 394-397

- Thank you for your recommendations. We have in particular used them to consider the performance/weights relationship, starting level issue, and new ranking system as mentioned above.
- We thank the author for their excellent and detailed critique.

Reviewer: 3

1. The main objective of the decision making of in this case (health service administration) should be addressed more specifically and clearly.

- We are unsure how to address this more succinctly in the text. The context specifically revolves around the challenges of hospital administrators in making informed decisions about multi-faceted interventions and projects. While we appreciate that this could be extended further, we were faced with word constraints and a limitation on the level of depth. We hope that we have addressed this sufficiently through the introduction in the abstract, through the background section, and through the rationale section on page 3.

2. How were these criteria selected? Did the authors do the expert validity assessment for the present criteria? What is the relevance of the criteria to the main decision-making objective? Is there independence between the criteria and no collinearity? Are there any other criteria to reach the main objective which have not been included yet?

- This is a good criticism that plagues MCDA as a discipline more generally. While there are many criteria available for MCDA, we were attempting to use only ones which were directly relevant to health services administrators. In doing so we consulted extensively with decision makers in the hospital and health service and found the best mix of parsimony and comprehensive coverage of issues. This is addressed in Methods, last paragraph of page 3.

3. The weights of the criteria were set beforehand by decision bodies, which might be too subjective and not objective.

- We agree that weightings are subjective, and this was the intention of the weighting component. Decision analysis requires priorities, and the weighting system was an explicit way of defining priorities. We have taken the feedback into consideration and changed the ranking system in order to account for this, now using an inverse weight system as detailed in the methods. This is addressed on pages 6-7 under 'Weighting and discounting for outcomes in future time periods'.

4. It is recommended that the author present the decision results in a "priority order" and explain how to apply the analysis results to subsequent actual decisions.

- We have, as mentioned above, taken this into account and designed a priority order ranking system for scores, pages 6-7 under 'Weighting and discounting for outcomes in future time periods'.

5. That would be better if the author can do the sensitivity analysis by the range of weights.

- We would ideally have included this information if there were no space constraints. Ultimately we were unable to conduct a sensitivity analysis, but the weighting method should explain the range of potential outcomes as a result of changes in weighted scores.
- We thank the reviewer for their comments and hope that we have sufficiently addressed their concerns.

VERSION 2 – REVIEW

REVIEWER	Jesper Bo Nielsen Department of Public Health University of Southern Denmark Denmark
REVIEW RETURNED	30-Dec-2018

GENERAL COMMENTS	Result section: To allow the reader to follow your calculations please add information on max scores for each of your 6 parameters. ROI for ES Pod: How did the authors get to this number (0.33)? Based on the available info, I could not
--

	Safety for EPI Center: Assuming a max value in this setting of 5, and not as described in the other example above 6? Line 9-11 in Discussion: A very pertinent remark. Might deserve 1-2 sentences more. Most focus tend to be given to the north-east quadrant, where you invest and gain, whereas the south-west quadrant is less often visited. But in an constrained economic situation we will need to look here as well. Last sentence above 'limitations in Discussion section: 'completing' should probably be corrected to 'competing'
--	--

REVIEWER	Jason C. Hsu National Cheng Kung University, Taiwan
REVIEW RETURNED	02-Jan-2019

GENERAL COMMENTS	This is an interesting, well-written paper about evaluation of health services administration by using MCDA method. The authors have answered all of my previous questions and have made the revision following my suggestions. I think the current version is good.
--

VERSION 2 – AUTHOR RESPONSE

Reviewer: 1

Result section:

To allow the reader to follow your calculations please add information on max scores for each of your 6 parameters.

- This has now been added to each segment of the results.

ROI for ES Pod: How did the authors get to this number (0.33)? Based on the available info, I could not

- Thank you for pointing this out. We had made a minor error in the discounting calculations of the ROI section. Once corrected, this also caused some minor changes to the main results in Table 6, which have been fixed.

Safety for EPI Center: Assuming a max value in this setting of 5, and not as described in the other example above 6?

- We have specified the max value as 5.

Line 9-11 in Discussion:

A very pertinent remark. Might deserve 1-2 sentences more. Most focus tend to be given to the north-east quadrant, where you invest and gain, whereas the south-west quadrant is less often visited. But in an constrained economic situation we will need to look here as well.

- We have added the following two sentences: “This tool can provide the rationale for disinvestment in projects occupying the top left quadrant, allowing funds to be better spent in the high-value sections. It can also help determine where to invest or disinvest in situations where financial constraints force an organisation to reduce benefits in order to cut costs.”

Last sentence above 'limitations in Discussion section: 'completing' should probably be corrected to 'competing'

- Thank you, amended.

- This review has been extremely helpful. Thanks again.

Reviewer: 3

This is an interesting, well-written paper about evaluation of health services administration by using MCDA method. The authors have answered all of my previous questions and have made the revision following my suggestions. I think the current version is good.

- Thank you for your feedback.